# NP-Engine: Empowering Optimization Reasoning in Large Language Models with Verifiable Synthetic NP Problems

## Abstract

Large Language Models (LLMs) have shown strong reasoning capabilities, with models like OpenAI's O-series and DeepSeek R1 excelling at tasks such as mathematics, coding, logic, and puzzles through Reinforcement Learning with Verifiable Rewards (RLVR). However, their ability to solve more complex optimization problems—particularly NP-hard tasks—remains underexplored. To bridge this gap, we propose NP-Engine, the first comprehensive framework for training and evaluating LLMs on NP-hard problems. NP-Engine covers 10 tasks across five domains, each equipped with (i) a controllable instance generator, (ii) a rule-based verifier, and (iii) a heuristic solver that provides approximate optimal solutions as ground truth. This generator-verifier-heuristic pipeline enables scalable and verifiable RLVR training under hierarchical difficulties. We also introduce NP-Bench, a benchmark derived from NP-Engine-Data, specifically designed to evaluate LLMs' ability to tackle NP-hard level reasoning problems, focusing not only on feasibility but also on solution quality. Additionally, we present Qwen2.5-7B-NP, a model trained via zero-RLVR with curriculum learning on Qwen2.5-7B-Instruct, which significantly outperforms GPT-4o on NP-Bench and achieves SOTA performance with the same model size. Beyond in-domain tasks, we demonstrate that RLVR training on NP-Engine-Data enables strong out-of-domain (OOD) generalization to reasoning tasks (logic, puzzles, math, and knowledge), as well as non-reasoning tasks such as instruction following. We also observe a scaling trend: increasing task diversity improves OOD generalization. These findings suggest that task-rich RLVR training is a promising direction for advancing LLM's reasoning ability, revealing new insights into the scaling laws of RLVR.

## 1 Introduction

Large Language Models (LLMs) have made significant advancements in complex reasoning tasks such as mathematics He et al. (2025), coding Liu & Zhang (2025), logic Xie et al. (2025), and puzzles Chen et al. (2025); Ma et al. (2024), showcasing the effectiveness of the Reinforcement Learning with Verifiable Rewards (RLVR) paradigm Xu et al. (2025b); Albalak et al. (2025). RLVR leverages high-quality, verifiable reward signals to guide model optimization, thereby enhancing LLMs' reasoning abilities OpenAI (2024); Guo et al. (2025); Google (2025); Anthropic (2025).

Despite significant advances, most existing research on reasoning in logic, mathematics, and puzzles focuses primarily on producing the "correct" answer, emphasizing solution accuracy. This emphasis overlooks solution quality, particularly in tasks that require not just *feasible* answers but *optimal* solutions. This gap highlights a crucial aspect of reasoning ability, referred to as **optimization reasoning** Li et al. (2025b).

To comprehensively evaluate the optimization capabilities of LLMs, we focus on utilizing LLMs to solve NP-hard problems, which involve complex combinatorial constraints and large problem spaces, posing significant optimization challenges. Since obtaining the optimal solution is computationally intractable in polynomial time, achieving better solutions requires the model to engage in

advanced reasoning, iterative trial-and-error, the generation of initial feasible solutions, and continuous self-reflection and optimization.

Recent works, such as NPHardEval Fan et al. (2024) and NPPC Yang et al. (2025b), have explored LLMs' performance on NP-hard problems. However, these studies are primarily focused on evaluation and suffer from limitations, such as insufficient control over difficulty levels and scalability or reliance on coarse metrics, which restrict their applicability in RLVR. Specifically, they do not offer the fine-grained optimal solutions necessary for generating high-quality reward signals, posing a significant challenge for integrating LLMs with RLVR in NP-level optimization tasks.

To address this gap, we propose NP-ENGINE, a unified framework for training and evaluating LLMs on NP-level optimization tasks using RLVR. NP-ENGINE includes 10 tasks across five domains, each featuring: (i) a scalable generator that produces instances with tunable difficulty, (ii) a rule-based verifier for automatic evaluation, and (iii) a heuristic algorithm that generates approximate optimal solutions. This generator-verifier-heuristic pipeline facilitates scalable RLVR training and allows for a fine-grained analysis of LLM performance.

To assess optimization capabilities, we introduce NP-BENCH, a benchmark that categorizes the 10 tasks into five primary categories, each containing 100 problems. Additionally, we design two metrics—Success Rate (SR) and Average Ratio (AR)—to evaluate the feasibility and optimality of the LLM's solutions. We also develop QWEN2.5-7B-NP, trained on Qwen2.5-7B-Instruct using zero-shot RLVR with curriculum learning strategies. QWEN2.5-7B-NP significantly outperforms GPT-4o on NP-BENCH, achieving SOTA SR across all LLMs and the highest AR among LLMs of the same size. Additionally, we evaluate the out-of-distribution (OOD) generalization of QWEN2.5-7B-NP on diverse reasoning tasks (logic, math, puzzles, knowledge) as well as non-reasoning tasks (instruction-following). Our results reveal that increasing task diversity enhances OOD performance, offering new insights into the scaling laws of RLVR for complex reasoning tasks. Our contributions are summarized as follows:

- We introduce NP-ENGINE, a scalable framework that generates near-infinite and hierarchically difficult NP-hard problems within the RLVR paradigm, empowering LLMs' optimization reasoning abilities. NP-ENGINE enables Qwen2.5-7B-Instruct to significantly outperform GPT-4o in optimization reasoning tasks using only 5K training examples.

- We propose NP-BENCH, a benchmark consisting of 10 NP-level tasks spanning five categories: Graph Clustering, Resource Scheduling, Graph Partitioning, Subset Selection, and Path Planning. NP-BENCH provides instances with varying difficulty levels and evaluates both the feasibility and quality of solutions.

- Through extensive experiments, we demonstrate that training on NP-ENGINE-DATA enables QWEN2.5-7B-NP to generalize to both reasoning and non-reasoning OOD tasks. We also observe a positive correlation between task diversity and cross-task generalization performance, offering new insights into the scaling behavior of RLVR-based training.

## 2 NP-ENGINE-DATA: THE OPTIMIZATION REASONING DATASET

### 2.1 NP PROBLEM CATEGORIES

As shown in Figure 1, NP-ENGINE-DATA comprises ten NP tasks of five main categories, including:

**Graph Clustering.** Select vertex sets under strict adjacency constraints to optimize structural objectives, requiring graph structural understanding and reasoning ability. Canonical problems include `Maximum Clique`, `Maximum Independent Set`, and `Graph Coloring`, focusing on dense-substructure clustering, mutual exclusivity, and chromatic feasibility.

**Resource Scheduling.** Assign activities to limited resources while avoiding conflicts and maximizing a global objective. The `Meeting Scheduling` task explores temporal feasibility, capacity limits, attendance constraints, and multi-constraint optimization.

**Graph Partitioning.** Partition a graph into (near-)equal parts while minimizing cut cost. `Balanced Minimum Bisection` balances partition constraints with inter-part edge weights, emphasizing cut minimization under cardinality constraints.

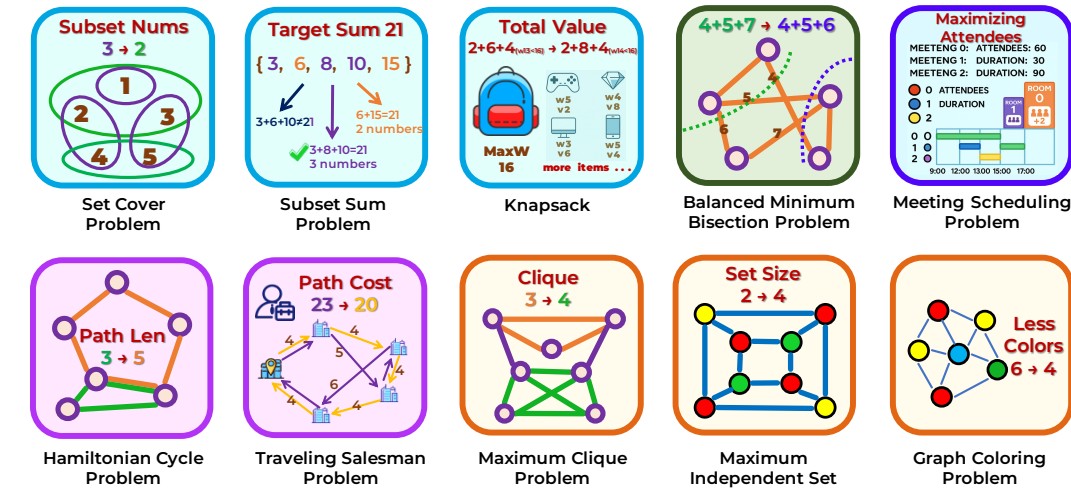

Figure 1: Overview of the NP-ENGINE-DATA: 10 NP-hard level tasks across five categories (graph clustering, resource scheduling, graph partitioning, subset selection, and path planning), designed to improve and evaluate optimization reasoning capabilities in LLMs.

**Subset Selection.** Choose subsets under combinatorial constraints to optimize coverage or sum/weight limits. Representative tasks such as `Subset Sum`, `Set Cover`, and `Knapsack` involve discrete feasibility checks and value–weight trade-offs.

**Path Planning.** Find short tours or feasible cycles visiting all nodes under specific distance metrics. The `Traveling Salesman Problem` and `Hamiltonian Cycle` test global optimization of permutations, path construction, and cycle feasibility.

## 2.2 NP-ENGINE-DATA CONSTRUCTION

We outline a four-stage data construction pipeline for NP-ENGINE-DATA: (I) Task Collection and Design; (II) Auto Task Generator and Verifier Development; (III) Heuristic Algorithm Construction; (IV) Task Difficulty Definition.

**Stage I: Task Collection and Design.** We curate 10 NP-Hard tasks requiring complex reasoning capabilities. Each task is scalable with custom generators to create additional NP-Hard instances, featuring optimal reasoning that integrates various reasoning skills. For example, finding the longest Hamiltonian circuit in a given graph.

**Stage II: Auto Task Generator and Verifier Development.** We equip the 10 tasks in NP-ENGINE-DATA with custom auto-generators for NP-Hard instances, enabling automatic data scaling for training and evaluation. Each task has a corresponding rule-based verifier, manually validated to assess the correctness of model outputs or provide rewards and penalties for the model's responses.

**Stage III: Heuristic Algorithm Construction.** We develop heuristic algorithms for each task to generate sub-optimal solutions, serving as an upper bound for evaluating LLM performance on NP-Hard tasks. These algorithms are efficient and scalable, enabling solution generation for large task instances. For example, for the Traveling Salesman Problem (TSP), we use the multi-start nearest neighbor heuristic to generate an initial shortest circuit, then optimize it through local search until no further improvement is made or a timeout occurs.

**Stage IV: Task Difficulty.** For each NP task, difficulty levels are determined by the size and complexity of problem instances. These parameters are controlled in the auto-generator to create instances across varying difficulty levels. For example, in TSP, we adjust the number of cities and path density to generate different levels of difficulty. We define three difficulty levels—Easy, Medium, and Hard—based on performance trends observed in the solution's success rate and quality across different parameter settings.

Table 1: Comparison of different reasoning benchmarks.

| Benchmark | Task Type | Tasks | Scalable | Verifier | Trainable |
|---|---|---|---|---|---|
| KOR-Bench Ma et al. (2024) | Knowledge | 125 | ✗ | ✓ | ✗ |
| NPR Wu et al. (2025) | Knowledge | 1 | ✗ | ✗ | ✗ |
| Logic-RL Xie et al. (2025) | Logic | 1 | ✓ | ✓ | ✓ |
| ZebraLogic Lin et al. (2025) | Logic | 1 | ✓ | ✓ | ✓ |
| SearchBench Borazjanizadeh et al. (2024) | Puzzle | 11 | ✓ | ✓ | ✗ |
| Enigmata Chen et al. (2025) | Puzzle | 36 | ✓ | ✓ | ✓ |
| NPHardEval Fan et al. (2024) | NP | 9 | ✗ | ✗ | ✗ |
| NPPC Yang et al. (2025b) | NP | 25 | ✓ | ✗ | ✗ |
| NP-BENCH | NP | 10 | ✓ | ✓ | ✓ |

## 2.3 NP-BENCH: THE OPTIMIZATION REASONING BENCHMARK

After constructing NP-ENGINE-DATA, we introduce NP-BENCH, a benchmark derived from NP-ENGINE-DATA for evaluating LLMs on NP-hard optimization reasoning tasks. NP-BENCH spans 10 tasks across five optimization categories. Table 1 compares NP-ENGINE-DATA with prior benchmarks. Unlike existing NP datasets, NP-ENGINE-DATA provides scalable instance generators and verifiers for all 10 tasks, enabling large-scale training and evaluation under controlled difficulty levels. Heuristic solvers are included to offer strong reference baselines and support automatic scoring. For each task, NP-BENCH offers 100 instances with high complexity (e.g., TSP instances with 45 to 55 cities), generated by task-specific generators to ensure diverse structures and constraints. We evaluate the solutions using two metrics: **Success Rate (SR)**, which measures the rate of feasible solutions, and **Average Ratio (AR)**, the mean quality ratio of the model's solution compared to a task-specific heuristic baseline, with infeasible cases scored as 0. This unified approach captures both feasibility and solution quality, providing a concise and comparable measure of LLMs' ability to solve NP-hard optimization problems.

## 3 NP-RL: THE TRAINING RECIPE

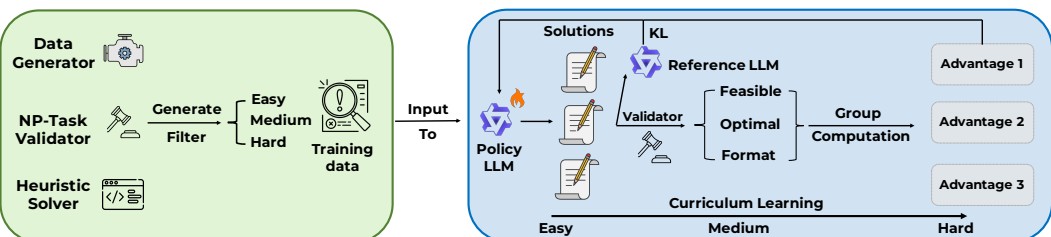

Figure 2: Overview of NP-ENGINE. The suite comprises ten NP-hard tasks spanning five categories (graph structure, scheduling, partitioning, subset selection, and tour planning). Each task is equipped with scalable instance generators and verifiers and is stratified into *Easy/Medium/Hard*, enabling RLVR curriculum training in LLMs.

Developing advanced optimization reasoning in large language models (LLMs) requires a well-structured training strategy. These optimization tasks must not only enable the model to find feasible solutions but also emphasize solution quality. Furthermore, when addressing multiple tasks, the model must exhibit versatile reasoning capabilities while avoiding overfitting to specific problem types. To address these challenges, our training framework is structured into three stages: (1) defining verifiable rewards for each task, (2) designing strategies to enhance optimization reasoning, and (3) applying multi-task learning to cultivate generalizable reasoning skills across diverse domains.

## 3.1 RL WITH VERIFIABLE REWARD

As shown in Figure 2, the reward serves as the guiding signal for the model to learn effective optimization strategies during the reinforcement learning process. To ensure the model not only

generates feasible solutions but also optimal solutions, our reward function consists of three main components: (1) format reward, to encourage deep reasoning thinking, (2) feasibility reward, which ensures that the generated solutions meet the problem constraints, and (3) optimality reward, which motivates the model to find the optimal solutions.

**Format Reward.** The format reward ensures that the model generates solutions in the correct format, adhering to the structure: "Please think step by step and output the chain of thought, and your response should end with: Answer: YOUR ANSWER". The format reward $R_{\text{format}}$ is defined as:

$$R_{\text{format}} = \begin{cases} 1, & \text{if format is correct} \\ -1, & \text{if format is incorrect} \end{cases}$$

**Feasibility Reward.** The feasibility reward ensures that the generated solutions satisfy problem constraints, such as finding a Hamiltonian cycle in the graph, which requires avoiding repeated nodes and invalid paths. The feasibility reward $R_{\text{feasibility}}$ is defined as:

$$R_{\text{feasibility}} = \begin{cases} R_{\text{optimal}}, & \text{if solution is valid} \\ -1.5, & \text{if solution is not valid} \end{cases}$$

**Optimal Reward.** The optimality reward encourages the model to find the best possible solution depending on the optimization goal. The optimality reward $R_{\text{optimal}}$ is defined as:

$$R_{\text{optimal}} = \begin{cases} \frac{M_s}{M_h}, & \text{if task target is maximum optimization} \\ \frac{M_h}{M_s}, & \text{if task target is minimum optimization} \end{cases} \in (0, 1]$$

where $M_s$ is the metric value of the solution calculated by NP-VALIDATOR, and $M_h$ is the metric value of the optimal solution generated by NP-HEURISTIC. The optimality reward is designed to be in the range $(0, 1]$, with higher values indicating better solutions. The overall reward $R$ is the sum of these components:

$$R = R_{\text{format}} + R_{\text{feasibility}}$$

### 3.2 Curriculum Learning Enhancing Optimization Capabilities.

Exposing models to overly complex tasks early in training can hinder the development of essential problem-solving skills, particularly in RLVR, where it leads to sparse reward signals. A key challenge is ensuring mastery of foundational skills before progressing to harder problems. To address this, we introduce Curriculum Learning, where the model first learns simpler concepts and gradually progresses to more complex ones. Unlike random data shuffling, curriculum learning incrementally increases task difficulty, ensuring a more structured learning process. The difficulty levels are hierarchically designed based on problem size, complexity, and constraints. Initially, the model trains on easier tasks, establishing a strong foundation. As training progresses, the model tackles more challenging tasks, building on prior knowledge and deepening its understanding of optimization reasoning. This approach promotes better model convergence and performance, as the model leverages previously learned knowledge to solve increasingly complex problems. By structuring the learning process in this way, we ensure that the model develops a solid understanding of basic reasoning, ultimately enhancing its ability to generalization across multiple NP tasks.

### 3.3 Multi-Stage RL Fostering Generalizable Reasoning Skills.

After achieving stable optimization reasoning abilities on a single task, we extend our approach to multi-task training, where the model is trained on all 10 tasks simultaneously during the RLVR process. Multi-task RL exposes the model to a wide variety of NP problem types, but the distinct reasoning approaches required for each task can create conflicts, hindering effective learning. To address this, we employ a multi-stage RL approach. In the first stage, we start with a single epoch of multi-task training to establish basic optimization reasoning abilities. Once these foundational skills are established, additional epochs are introduced to further enhance the model's capabilities, progressing through up to three stages. This staged approach allows the model to gradually adapt to the complexities of different tasks, ensuring effective learning and generalization across a broad range of NP problems. As a result, the model's optimization reasoning performance improves, while its ability to perform "deep thinking" during RLVR training significantly enhances its generalization to tackle other reasoning tasks.

Table 2: Performance of reasoning LLMs, general LLMs, and our trained LLMs on NP-BENCH.

| Model | Graph | | Schedule | | Partition | | Selection | | Planning | | Overall | |
|---|---|---|---|---|---|---|---|---|---|---|---|---|
| | SR | AR | SR | AR | SR | AR | SR | AR | SR | AR | SR | AR |
| *Proprietary LLMs* | | | | | | | | | | | | |
| DS-V3.1-Thinking | 86.0 | 78.2 | **99.0** | 91.4 | 98.0 | **77.1** | **99.3** | **98.5** | 61.9 | 54.3 | 88.8 | **79.9** |
| gpt-o3 | **97.0** | **86.4** | **99.0** | **94.5** | **100.0** | 51.4 | 87.4 | 87.3 | 74.0 | **65.1** | 91.5 | 76.9 |
| Qwen3-235B-Thinking | 66.7 | 62.9 | 95.0 | 93.0 | **100.0** | 55.8 | 98.0 | 97.1 | 52.0 | 44.8 | 82.3 | 70.7 |
| gpt-4o-2024-08-06 | 64.7 | 29.3 | 79.0 | 59.8 | **100.0** | 53.0 | 14.7 | 9.3 | 52.0 | 29.6 | 62.1 | 36.2 |
| *Open-Source LLMs* | | | | | | | | | | | | |
| Qwen3-32B | 44.7 | 39.3 | 94.0 | 93.9 | 99.0 | 52.6 | 94.1 | 91.4 | 21.6 | 11.2 | 70.7 | 57.6 |
| Qwen3-8B | 22.7 | 16.8 | 78.0 | 75.3 | 98.0 | 51.0 | 86.0 | 82.6 | 3.0 | 1.2 | 57.5 | 45.4 |
| DS-R1-Qwen-32B | 23.3 | 18.1 | 49.0 | 45.1 | 96.0 | 48.6 | 85.7 | 79.7 | 15.4 | 7.9 | 53.9 | 39.9 |
| DS-R1-Qwen-14B | 18.0 | 13.4 | 52.0 | 51.7 | 32.0 | 16.2 | 67.3 | 63.4 | 4.5 | 1.4 | 34.8 | 29.2 |
| Qwen2.5-72B | 34.7 | 15.2 | 59.0 | 58.5 | 90.0 | 39.5 | 27.0 | 17.4 | 6.5 | 2.1 | 43.4 | 26.5 |
| Qwen2.5-32B | 35.3 | 15.2 | 15.0 | 12.8 | **100.0** | 51.7 | 32.0 | 22.4 | 23.5 | 6.7 | 41.2 | 21.8 |
| Qwen2.5-14B | 30.0 | 11.5 | 21.0 | 15.8 | 89.0 | 44.3 | 23.3 | 12.7 | 17.5 | 4.9 | 36.2 | 17.8 |
| InternLM3-8b | 15.0 | 3.6 | 20.0 | 9.5 | 86.0 | 43.3 | 41.3 | 23.7 | 16.0 | 4.1 | 35.7 | 16.8 |
| LLama3.1-8B | 23.0 | 8.0 | 9.0 | 7.8 | 0.0 | 0.0 | 28.7 | 11.4 | 15.0 | 1.8 | 15.1 | 5.8 |
| Qwen2.5-3B | 7.7 | 2.9 | 17.0 | 5.0 | 6.0 | 2.2 | 23.0 | 10.7 | 15.5 | 3.7 | 13.8 | 4.9 |
| DS-R1-Qwen-7B | 6.3 | 1.9 | 1.0 | 0.9 | 2.0 | 1.0 | 13.7 | 8.9 | 0.5 | 0.1 | 4.7 | 2.5 |
| Qwen2.5-7B | 11.0 | 3.1 | 40.0 | 19.8 | 67.0 | 34.0 | 26.7 | 15.2 | 3.5 | 1.0 | 29.6 | 14.6 |
| QWEN2.5-7B-NP | 89.7 | 27.8 | 85.0 | 43.5 | 99.0 | 53.8 | 93.7 | 79.1 | **98.2** | 28.9 | **93.1** | 46.6 |
| | +78.7 | +24.7 | +45.0 | +23.7 | +32.0 | +19.8 | +67.0 | +63.9 | +94.7 | +27.9 | +63.5 | +32.0 |

## 4 EXPERIMENT

### 4.1 EXPERIMENTAL SETUP

We evaluate the performance of our proposed method on a range of benchmark datasets spanning multiple domains, including NP-BENCH and five out-of-domain benchmarks. These benchmarks are categorized as follows: four **reasoning** benchmarks—(I) **Logic Reasoning**: KOR-Bench; (II) **Math Reasoning**: MATH500 and OlympiadBench; (III) **Knowledge Reasoning**: GPQA_Diamond—and one **Non-Reasoning Task**: IFEVAL, which includes factual and alignment-based instruction-following questions. All experiments are conducted using the OpenCompass Contributors (2023) framework.

Our QWEN2.5-7B-NP is directly trained from Qwen2.5-7B-Instruct-1M Yang et al. (2025a), as we found that applying RLVR to Qwen2.5-7B-Instruct often leads to poor instruction adherence and formatting issues. The detailed training setup is provided in the Appendix.

### 4.2 EVALUATION RESULTS ON NP-BENCH

As shown in Table 2, we evaluate the performance of LLMs on NP-BENCH using Success Rate (SR) and Average Ratio (AR) as metrics. Our model, QWEN2.5-7B-NP, shows significant improvements over the baseline across all sub-tasks. Notably, the overall SR increases from 29.6 to 93.1, achieving state-of-the-art performance on all models, and the AR rises from 14.6 to 46.6, maintaining state-of-the-art performance for the same model size. The gains are especially pronounced in tasks with complex structures and constraints, such as *Graph* and *Selection*, where both SR and AR improve by several-fold. *Scheduling* tasks, which require balancing temporal feasibility and capacity constraints, also exhibit large relative gains. Even on *Partition*, where the baseline is already strong, QWEN2.5-7B-NP delivers consistent gains, while in *Planning* it achieves near-perfect accuracy.

These results emphasize the significant impact of zero-RLVR on enhancing in-domain optimization reasoning abilities, particularly for tasks like graph-structured search and discrete selection. Additionally, tasks involving scheduling and planning show considerable improvements, demonstrating the model's ability to handle complex constraints and optimization requirements with task-specific reinforcement learning.

### 4.2.1 EVALUATION RESULTS ON OUT-OF-DOMAIN (OOD) BENCHMARKS

As shown in Table 3, we evaluate out-of-domain (OOD) tasks across five categories: four reasoning tasks—*Logic* (KORBench), *Math* (Math500 and OlympiadBench), *Knowledge*

Table 3: Performance on out-of-domain benchmarks, including both reasoning and non-reasoning tasks, demonstrates that RLVR training on NP-ENGINE-DATA generalizes effectively.

| Model | Logic | Math | | Knowledge | Instruction | Average |
|-------|-------|------|---|-----------|-------------|---------|
| | KORBench | Math500 | OlpBench | GPQA_diamond | IFEval | |
| LLama3.1-8b | 44.5 | 48.6 | 17.7 | 22.7 | 69.3 | 40.6 |
| Qwen2.5-3B | 36.6 | 67.2 | 29.7 | 30.3 | 59.1 | 44.6 |
| InternLM3-8B | 40.7 | 78.2 | 25.1 | 36.9 | 72.5 | 50.7 |
| Qwen2.5-14B | 50.6 | 80.0 | 45.1 | 41.9 | 81.6 | 59.8 |
| Qwen2.5-32B | **56.2** | 82.8 | 48.8 | 43.9 | 79.5 | 62.3 |
| Qwen2.5-72B | 53.0 | **84.2** | **49.1** | **46.0** | **84.3** | **63.3** |
| Qwen2.5-7B | 42.9 | 72.4 | 30.0 | 33.3 | 73.5 | 50.4 |
| QWEN2.5-7B-NP | 44.1 (+1.2) | 74.6 (+2.2) | 32.1 (+2.1) | 37.4 (+4.1) | 79.6 (+6.1) | 53.6 (+3.2) |

Table 4: Comparison of different data proportions for easy (E), medium (M), and hard (H) from NP-ENGINE-DATA during RLVR, as well as curriculum learning (CL) strategies.

| Data Proportion | CL | In Domain | | Out of Domain | | | | | |
|-----------------|----|-----------|---|---------------|---|---|---|---|---|
| | | SR | AR | KB | Math500 | OB | GPQA | IF | Avg |
| Qwen2.5-7B (Base) | ✗ | 4.0 | 1.9 | 42.9 | 72.4 | 30.0 | 33.3 | 73.5 | 50.4 |
| Qwen2.5-7B (w/o GT) | ✗ | 90.0 | 25.6 | 43.4 | 73.0 | 30.5 | 29.3 | 78.3 | 50.9 |
| E:M:H=1:4:5 | ✗ | 99.0 | 28.1 | 42.9 | 73.8 | **31.1** | 35.4 | 77.9 | 52.2 |
| | ✓ | 98.0 | 28.2 | 43.4 | 73.6 | 30.8 | **37.9** | 78.5 | 52.8 |
| E:M:H=1:1:1 | ✗ | 97.0 | 26.9 | 42.3 | 74.2 | 29.6 | 32.3 | 78.8 | 51.5 |
| | ✓ | 98.0 | 27.1 | **44.6** | **74.8** | 29.4 | 32.8 | **79.3** | 52.2 |
| E:M:H=5:4:1 | ✗ | **100.0** | 28.2 | 44.2 | 74.4 | 30.4 | 31.3 | 78.5 | 51.8 |
| | ✓ | **100.0** | **29.0** | 44.3 | 74.4 | **31.1** | 35.9 | 78.6 | **52.9** |

(GPQA_diamond)—and one non-reasoning task, *Instruction Following* (IFEval). Our QWEN2.5-7B-NP demonstrates strong generalization capabilities compared to the baseline, with the overall average score increasing from **50.4** to **53.6**. Improvements are observed across all categories, with reasoning benchmarks such as Logic, Math500, and OlympiadBench showing consistent gains of around 2–3 points, and GPQA_diamond improving by over 12%. Even on non-reasoning tasks like IFEval, performance rises by more than 8%.

Notably, the RLVR approach on NP-ENGINE-DATA leads to significant improvements in out-of-domain reasoning tasks (logic, math, knowledge), demonstrating that zero-RLVR enhances the model's generalization ability by fostering deeper reasoning for complex NP-hard optimization tasks, rather than overfitting to in-domain data. Additionally, our reward design improves instruction-following performance, yielding enhancements in non-reasoning tasks. These results highlight the effectiveness of the NP-ENGINE framework in enhancing both reasoning and non-reasoning capabilities.

## 4.3 ABLATION STUDY

### 4.3.1 CURRICULUM LEARNING AND DATA PROPORTION

As shown in Table 4, we investigate the impact of different data proportions—easy (E), medium (M), and hard (H)—on RLVR training, along with the role of curriculum learning (CL) strategies. The experiments focus on the TSP problem to better summarize the rules. We compare several configurations, including the baseline model and a variant without NP-HEURISTIC to provide accurate ground-truth (GT) signals in the reward signal, as well as different data ratios (E:M:H). In terms of in-domain performance, all RLVR configurations show improvements. Even the `Qwen2.5-7B` `(w/o GT)` model achieves noticeable gains, with SR rising from 4.0 to 90.0 and AR from 1.9 to 25.6. The introduction of curriculum learning (CL) results in further gains across all data proportions. The best performance is achieved with the *E:M:H=5:4:1* configuration, which achieves

Table 5: Performance on out-of-domain benchmarks, with increasing task scale from NP-ENGINE-DATA during RLVR training.

| Task Number | Logic | Math | | Knowledge | Instruction | Average |
|---|---|---|---|---|---|---|
| | KORBench | Math500 | OlpBench | GPQA | IFEval | |
| Qwen2.5-7B | 42.9 | 72.4 | 30.0 | 33.3 | 73.5 | 50.4 |
| +3 Tasks | 43.8 (+0.9) | 74.8 (+2.4) | 30.5 (+0.5) | 37.4 (+4.1) | 78.5 (+5.0) | 53.0 (+2.6) |
| +5 Tasks | 44.2 (+1.3) | 73.2 (+0.8) | 31.9 (+1.9) | 37.4 (+4.1) | 78.0 (+4.5) | 52.9 (+2.5) |
| +7 Tasks | 44.2 (+1.3) | 76.2 (+3.8) | 30.4 (+0.4) | 35.9 (+2.6) | 78.8 (+5.3) | 53.1 (+2.7) |
| +ALL Tasks | 44.1 (+1.2) | 74.6 (+2.2) | 32.1 (+2.1) | 37.4 (+4.1) | 79.6 (+6.1) | 53.6 (+3.2) |

an average AR of 29.0, outperforming other configurations. For out-of-domain tasks, particularly in the *Math500* and *GPQA* benchmarks, the E:M:H=5:4:1 configuration demonstrates superior generalization with an OOD average of 52.9. The inclusion of curriculum learning stabilizes performance and enhances the model's ability to generalize across both reasoning-heavy and instruction-following tasks.

Overall, these experiments highlight the importance of data proportion in RLVR, particularly the need for a larger proportion of easy tasks to build a strong foundation before tackling more complex problems. Curriculum learning further enhances this process, improving both in-domain and out-of-domain generalization capabilities.

### 4.3.2 MULTI-STAGE RL RECIPE

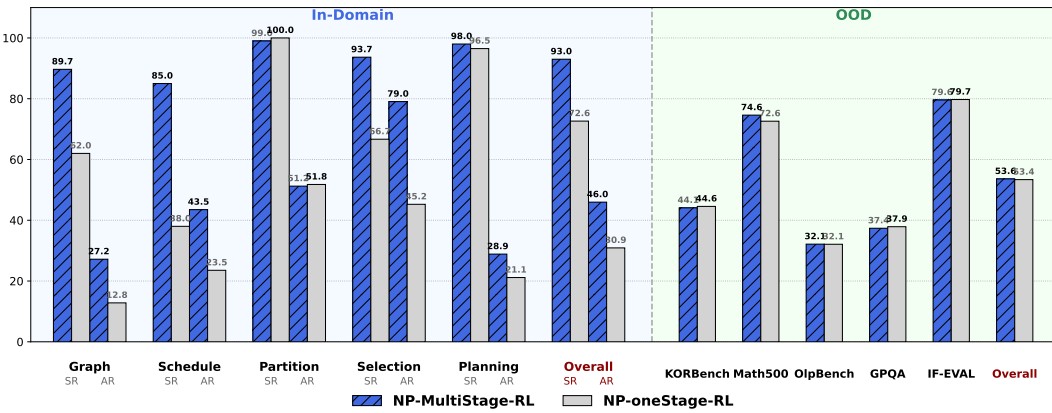

Figure 3: Comparison of RL training strategies during multi-task training, with performance evaluated on both in-domain and out-of-domain benchmarks.

As shown in Figure 3, compared to OneStage-RL, which uses NP-ENGINE-DATA in a single epoch, MultiStage-RL divides NP-ENGINE-DATA into multiple epochs. The NP-MultiStage-RL strategy consistently outperforms NP-OneStage-RL across all sub-tasks in both in-domain and out-of-domain (OOD) settings. In the in-domain tasks, MultiStage RL demonstrates significant improvements in Success Rate (SR) and Average Ratio (AR) across all tasks. For example, in *Graph*, SR increases from 62.0 (OneStage) to 89.7 (MultiStage), and in *Selection*, SR rises from 56.6 to 93.7, marking a substantial gain. The overall SR for MultiStage RL reaches 93.0, surpassing OneStage RL's 72.6, underscoring its effectiveness in enhancing in-domain performance. In the OOD tasks, MultiStage RL also outperforms OneStage RL, with an overall SR of 53.6 compared to OneStage RL's 53.4. These results demonstrate that MultiStage RL significantly improves performance in both in-domain and OOD settings. By multi-stage adapting to the complexities of different tasks, MultiStage RL ensures effective learning and generalization across a broad range of NP problems.

### 4.3.3 SCALING UP TASKS ON NP-ENGINE-DATA

The results in Table 5 demonstrate the effect of task scaling on NP-ENGINE-DATA, with the model trained with data from **all tasks** achieving the highest average score of 53.31 across all NP tasks.

This underscores the advantages of a more comprehensive training approach that spans a wider variety of tasks. The Model trained with data from **7 tasks** performs particularly well on tasks like `Math500` (76.2) and `IFEval` (78.84), excelling in mathematical and instruction-following tasks. In comparison, **fewer tasks** as training data yield slightly lower scores in these areas. Models trained on only **3 tasks** show greater limitations on complex tasks, like `KORBench` (43.84) and `Math500` (74.8). These results emphasize that training on diverse tasks is crucial for achieving optimal performance across reasoning tasks. In conclusion, incorporating a broader task range enhances the model's generalization ability, providing valuable insights into the scaling laws of RLVR.

## 5 RELATED WORK

**Reinforcement Learning with Verifiable Rewards (RLVR).** As reinforcement learning (RL) becomes an increasingly important tool for enhancing the reasoning capabilities of LLMs, Reinforcement Learning with Verifiable Rewards (RLVR) has emerged as a compelling alternative to Reinforcement Learning with Human Feedback (RLHF). Unlike RLHF, which relies on pretrained reward models and subjective human annotations, RLVR utilizes objective, automatically verifiable outcomes to provide reliable supervision Seed et al. (2025); Guo et al. (2025); Team et al. (2025). Recent models exemplify this paradigm shift: DeepSeek-R1 Guo et al. (2025) improves long-chain reasoning and self-verification through RLVR, while Kimi K1.5 Team et al. (2025) achieves strong performance with long-context training and streamlined policy optimization—without depending on complex value models. The ecosystem supporting RLVR is rapidly maturing. High-quality math corpora with verifiable solutions He et al. (2025); Albalak et al. (2025), structured coding corpora with graded difficulty and reward pipelines Liu & Zhang (2025); Xu et al. (2025b), and procedurally generated puzzle-style datasets with algorithmic verification Xie et al. (2025); Chen et al. (2025); Li et al. (2025a) are now available. Notably, NP problems are inherently verifiable and offer controllable difficulty settings Fan et al. (2024); Yang et al. (2025b), making them well-suited for RLVR-based training. However, most prior RLVR efforts have focused on math, coding, logic, or puzzles, leaving the broader class of NP-hard problems underexplored.

**Optimization Reasoning with LLMs.** Various benchmarks have been proposed to evaluate LLMs' reasoning capabilities across different domains, including mathematical Glazer et al. (2024), logical Xie et al. (2025), puzzle Chen et al. (2025), and programming reasoning Xu et al. (2025a). These tasks typically involve binary answer validation (e.g., True or False), which primarily assess deductive or symbolic reasoning. In contrast, optimization reasoning presents a fundamentally different challenge: it requires models to generate not only feasible solutions but also solutions that are as optimal as possible. Previous work on NP tasks has faced challenges in trainability Fan et al. (2024); Yang et al. (2025b). Despite its significance, optimization reasoning has been underexplored in RLVR-based LLM training. Our work addresses this gap by focusing on NP-class problems. We propose NP-ENGINE, a unified framework for data generation, optimal solution annotation, RLVR training, and evaluation, which empowers LLMs with optimization reasoning capabilities.

## 6 CONCLUSION

In this work, we present NP-ENGINE, the first comprehensive framework for enabling LLMs to solve NP-hard optimization reasoning problems. By combining scalable instance generation, verifiable rule-based evaluation, and heuristic solvers to provide precise reward signals, NP-ENGINE supports effective RLVR-based training on complex optimization tasks. We also introduce NP-BENCH, a benchmark covering 10 diverse NP-level tasks across five primary optimization reasoning domains, and propose QWEN2.5-7B-NP, an RLVR-trained model that significantly outperforms GPT-4o on NP-BENCH. Our experiments demonstrate that curriculum learning strategies and multi-stage RL training substantially enhance LLMs' optimization reasoning capabilities. Furthermore, we observe a strong correlation between task diversity and generalization performance, offering insights into the scaling laws for RLVR-based training in complex reasoning domains. We hope this work lays a foundation for future research on integrating LLMs with optimization reasoning and highlights task-rich RLVR training as a promising direction for advancing LLM reasoning in complex optimization tasks. Additionally, our findings reveal new insights into the scaling laws of RLVR.

# REPRODUCIBILITY STATEMENT

We adhere to the reproducibility guidelines outlined in the ICLR 2026 author guidelines. All data, code, and checkpoints will be made publicly available to facilitate the reproduction of our results at the earliest opportunity.

## ETHICS STATEMENT

The NP-ENGINE-DATA dataset was constructed using NP-ENGINE, a rule-based NP data construction environment, ensuring that no ethical risks are involved.

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

# A    APPENDIX

## A.1    USE OF LARGE LANGUAGE MODELS

Large Language Models are used for grammar check and polishing in this paper.

## A.2    LIMITATION

Due to computational resource constraints, we have conducted experiments using the Qwen2.5-7B-Instruct model. Larger models, such as those with 14B or 32B parameters, have not been trained, and the performance of these more powerful models, starting from a bigger model size, remains unexplored. Additionally, designing individual NP tasks for RLVR training requires meticulous attention to various aspects, including problem definition, validation script development, heuristic algorithm design, and difficulty level calibration. As a result, we have currently designed 10 tasks. Scaling to larger models and incorporating additional tasks remain avenues for future exploration.

## A.3    DETAILED TRAINING SETTING

The training experiment utilizes the verl framework, employing the GRPO algorithm for fine-tuning the Qwen2.5-7B-Instruct-1M model. Training is performed on 8 A800 GPUs with a batch size of 8 for both training and validation. The maximum prompt length is set to 20,000 tokens, and the response length is capped at 4,096 tokens. Key hyperparameters include a learning rate of $4 \times 10^{-7}$, with a mini-batch size of 256 and a micro-batch size of 64 for PPO updates. KL loss regularization, with a coefficient of 0.001 ($\text{KL}_{\text{coef}}$), is applied to stabilize training. The model is trained for 3 epochs to ensure convergence.

## A.4    NP-HARD TASKS

### A.4.1    SET COVER

The task is to solve the classical Set Cover Problem. Given a universal set $U$ and a collection of subsets $S \subseteq 2^U$, the goal is to find the smallest possible sub-collection of $S$ whose union equals $U$. In other words, we aim to select the minimum number of subsets such that every element in $U$ is contained in at least one of the selected subsets. If no such selection exists, the answer should be "Impossible". The solution is represented as a list of subset indices corresponding to the chosen sub-collection.

For example, given $U = \{0, 1, 2, 3, 4, 5\}$ and

$$S = \{\ 0 : \{0, 1, 2\},\ 1 : \{2, 3\},\ 2 : \{0, 4\},\ 3 : \{3, 4, 5\},\ 4 : \{1, 2, 5\}\ \},$$

a valid minimum cover is $[0, 3, 4]$, since the union of these subsets is equal to $U$.

The difficulty of the problem instances is categorized based on the size of the universe $|U|$, the number of subsets $|S|$, and the relative subset size (controlled by the parameter subset_size_factor):

- **Easy**:
  - $|U| \in [10, 20]$, $|S| \in [5, 10]$, subset size factor = 0.4
  - Small universe and relatively large subsets, making coverage straightforward.
- **Medium**:
  - $|U| \in [20, 25]$, $|S| \in [10, 15]$, subset size factor = 0.4
  - Moderate universe size and subset count, requiring careful selection.
- **Hard**:
  - $|U| \in [25, 30]$, $|S| \in [15, 25]$, subset size factor = 0.4
  - Larger universe with more subsets, increasing combinatorial complexity.
- **Benchmark**:
  - $|U| \in [30, 40]$, $|S| \in [20, 30]$, subset size factor = 0.4
  - The most challenging setting, with the largest universes and dense subset collections.

### A.4.2 SUBSET SUM

The Subset Sum Problem asks whether a subset of integers sums up to a given target value $T$. In this variation, the objective is not only to reach the target sum, but also to maximize the number of elements used in the subset.

Formally, given a set of integers $\{a_0, a_1, \ldots, a_{n-1}\}$ and a target $T$, the task is to find an index set $I \subseteq \{0, 1, \ldots, n-1\}$ such that

$$\sum_{i \in I} a_i = T,$$

and among all valid solutions, the chosen $I$ maximizes $|I|$. If multiple such subsets exist, any of them is acceptable. The submission format requires returning the ordered list of indices, e.g., $[0, 1, 4]$.

For example, given

$$T = 10, \quad \text{numbers} = \{0 : 2,\ 1 : 3,\ 2 : 7,\ 3 : 8,\ 4 : 5\},$$

a valid solution is $[0, 1, 4]$, since $2 + 3 + 5 = 10$, and the subset uses three elements, which is maximal.

The difficulty of generated problem instances is categorized according to the number of integers available ($|\text{numbers}|$), the typical size of the optimal solution ($|I|$), and the range of integer values:

- **Easy**:
    - Total numbers $\in [5, 10]$, solution size $\in [4, 8]$, values in $[1, 5]$.
    - Small input with low values, ensuring frequent feasible solutions.
- **Medium**:
    - Total numbers $\in [8, 12]$, solution size $\in [4, 8]$, values in $[1, 10]$.
    - Moderate instance size and range, requiring more careful subset selection.
- **Hard**:
    - Total numbers $\in [12, 15]$, solution size $\in [8, 12]$, values in $[1, 15]$.
    - Larger solution sizes and wider value ranges increase combinatorial difficulty.
- **Benchmark**:
    - Total numbers $\in [15, 20]$, solution size $\in [10, 15]$, values in $[1, 15]$.
    - The most challenging setting, with large search space and dense feasible solutions.

### A.4.3 KNAPSACK

The Knapsack Problem requires selecting a subset of items to maximize the total value without exceeding a weight capacity. Formally, given a set of items $\{(w_i, v_i)\}_{i=0}^{n-1}$, each with weight $w_i$ and value $v_i$, and a knapsack capacity $W$, the goal is to find an index set $I \subseteq \{0, 1, \ldots, n-1\}$ such that

$$\sum_{i \in I} w_i \leq W, \quad \text{and} \quad \sum_{i \in I} v_i$$

is maximized. The submission format requires returning the ordered list of chosen item IDs in square brackets, e.g., $[0, 2, 3]$.

For example, with

$$W = 20, \quad \text{items} = \{0 : (3, 4),\ 1 : (4, 5),\ 2 : (7, 10),\ 3 : (8, 11)\},$$

a valid optimal solution is $[0, 2, 3]$, achieving total weight $18 \leq 20$ and total value 25.

The problem instances are categorized into four difficulty levels, determined by the number of items, their weight/value ranges, and the relative knapsack capacity:

- **Easy**:
    - $6 \leq |I^*| \leq 10$ (solution items), total items $\approx 15$–$25$.
    - Weights in $[5, 25]$, value-to-weight ratio in $[1.8, 2.5]$.

- Capacity is 1.1–1.4 times the total weight of the solution items.

- **Medium**:
  - $8 \leq |I^*| \leq 12$, total items $\approx$ 25–35.
  - Weights in $[20, 80]$, value-to-weight ratio in $[1.5, 2.0]$.
  - Capacity is 1.05–1.25 times the total solution weight.

- **Hard**:
  - $15 \leq |I^*| \leq 25$, total items $\approx$ 35–60.
  - Weights in $[50, 200]$, value-to-weight ratio in $[1.2, 1.6]$.
  - Capacity is 1.02–1.15 times the total solution weight.

- **Benchmark**:
  - $25 \leq |I^*| \leq 35$, total items $\approx$ 55–80.
  - Weights in $[50, 200]$, value-to-weight ratio in $[1.2, 1.6]$.
  - Capacity is 1.02–1.15 times the total solution weight.
  - The most challenging setting, with many items and tight capacity.

### A.4.4    BALANCED MINIMUM BISECTION

The Balanced Minimum Bisection Problem requires partitioning a weighted undirected graph $G = (V, E)$ into two disjoint subsets of nearly equal size (differing by at most one vertex) such that the sum of the weights of edges crossing the cut is minimized. Unlike the classic Minimum Cut Problem, this task includes a balance constraint: both partitions must contain approximately the same number of vertices.

Formally, let $V$ be divided into $V_1$ and $V_2$ such that $V_1 \cap V_2 = \emptyset$, $V_1 \cup V_2 = V$, and $\left||V_1| - |V_2|\right| \leq 1$. The objective is to minimize

$$\sum_{\substack{u \in V_1, v \in V_2 \\ (u,v) \in E}} w(u, v),$$

where $w(u, v)$ is the edge weight. The solution format specifies the two subsets explicitly, e.g., $[[0, 1, 2], [3, 4, 5]]$.

For example, consider the input graph:

$$0 : \{1 : 3, 2 : 1\}, \quad 1 : \{0 : 3, 2 : 2, 3 : 2\}, \quad 2 : \{0 : 1, 1 : 2, 3 : 3\}, \quad 3 : \{1 : 2, 2 : 3\}.$$

A valid optimal balanced bisection is $[[0, 1], [2, 3]]$.

The difficulty of generated instances is determined by the number of nodes, the structural complexity of the graph, and the noise level:

- **Easy**:
  - $|V| \approx 30$.
  - Graphs have clear community structures with dense intra-community edges and sparse inter-community connections, with small noise ($\approx 0.1$).
  - Balanced cuts are relatively easy to identify.

- **Medium**:
  - $|V| \approx 42$.
  - Graphs exhibit fuzzier community boundaries and more inter-community edges, with moderate noise ($\approx 0.15$).
  - Increases difficulty by reducing the clarity of the optimal partition.

- **Hard**:
  - $|V| \approx 45$.
  - Graphs are generated with deceptive structures, including "traitor" nodes and reinforced communities.
  - Noise level around 0.1, making near-optimal but incorrect cuts more likely.

- **Benchmark**:
  - $|V| \approx 50$.
  - Graphs include reinforced "hell mode" structures, traitor nodes, and low noise ($\approx 0.02$).
  - The most challenging setting, with multiple plausible partitions and high combinatorial complexity.

### A.4.5 Meeting Scheduling Problem

The Meeting Scheduling Problem (MSP) aims to assign meetings to rooms and times in order to maximize total attendee participation, subject to availability and capacity constraints. Each meeting requires a set of attendees and a duration, each attendee has availability intervals, and each room has a capacity. A feasible solution must assign to each scheduled meeting a start time and a room such that:

- All required attendees are available for the entire duration.
- The room has sufficient capacity for all attendees.
- No attendee or room is scheduled for overlapping meetings.

If a meeting cannot be scheduled under these constraints, it is omitted. The solution is expressed as an ordered list of tuples (meeting_id, room_id, start_time), sorted by start time.

For example, given the input

$$\text{meetings} = \{0 : ([0, 1, 2], 60), \ 1 : ([1, 3], 30), \ 2 : ([0, 2, 3], 90)\},$$

$$\text{availability} = \{0 : [(900, 1700)], \ 1 : [(900, 1200), (1300, 1700)], \ 2 : [(900, 1700)], \ 3 : [(1000, 1400)]\},$$

$$\text{rooms} = \{0 : 5, \ 1 : 3\},$$

a valid schedule is

$$[(0, 0, 900), \ (1, 1, 1000), \ (2, 0, 1020)],$$

which yields a total of 8 attendee participations.

The difficulty of generated MSP instances depends on the number of meetings, attendees, rooms, and fragmentation of availability:

- **Easy**:
  - 4–5 meetings, 3–5 attendees, 3–4 rooms.
  - At most 3 attendees per meeting.
  - Availability mostly continuous within the working day.
- **Medium**:
  - 5–6 meetings, 4–6 attendees, 4–5 rooms.
  - At most 4 attendees per meeting.
  - Some attendees have fragmented availability (e.g., lunch breaks).
- **Hard**:
  - 6–7 meetings, 5–7 attendees, 5–6 rooms.
  - At most 4 attendees per meeting.
  - Heavier overlap among meetings and tighter room capacities.
- **Benchmark**:
  - 8–10 meetings, 7–9 attendees, 6–7 rooms.
  - At most 5 attendees per meeting.
  - The most challenging setting, with dense scheduling conflicts and fragmented availability.

### A.4.6 HAMILTONIAN CYCLE

The task is to find a Hamiltonian circuit in a given graph $G$, which is a path that visits every vertex exactly once and returns to the starting point. The goal is to maximize the number of vertices included in the Hamiltonian circuit. The process starts with a random vertex and finds a small valid subgraph, then iteratively expands the subgraph while ensuring it remains valid, continuing until the largest possible Hamiltonian circuit is found.

The problem is categorized into four difficulty levels based on the number of vertices and edge density:

- **Easy**:
  - $|V| \in [15, 20]$, $\rho = 0.2$
  - Small graph with sparse edges.
- **Medium**:
  - $|V| \in [20, 30]$, $\rho = 0.3$
  - Moderate graph size with moderate connectivity.
- **Hard**:
  - $|V| \in [30, 40]$, $\rho = 0.4$
  - Larger graph with denser edges, increasing difficulty.
- **Benchmark**:
  - $|V| \in [40, 50]$, $\rho = 0.5$
  - The most challenging, with the largest and densest graph.

### A.4.7 TRAVELING SALESMAN PROBLEM

The Traveling Salesman Problem (TSP) is a classical combinatorial optimization problem. Given a set of cities and pairwise distances, the objective is to find the shortest possible tour that:

- Starts and ends at the same city.
- Visits each city exactly once in between.

The solution is expressed as a route $[c_0, c_1, \ldots, c_{n-1}, c_0]$, where $c_0$ is the starting city and each city appears exactly once except for the repetition of $c_0$ at the end.

For example, given the distance dictionary

$0 : \{1 : 10, 2 : 15, 3 : 20\}, \quad 1 : \{0 : 10, 2 : 35, 3 : 25\}, \quad 2 : \{0 : 15, 1 : 35, 3 : 30\}, \quad 3 : \{0 : 20, 1 : 25, 2 : 30\},$

a valid optimal solution is

$$[0, 1, 3, 2, 0].$$

The difficulty of generated TSP instances is determined primarily by the number of cities:

- **Easy**: 10–20 cities.
- **Medium**: 20–30 cities.
- **Hard**: 35–45 cities.
- **Benchmark**: 45–55 cities.

All instances are generated with symmetric distance matrices, with distances sampled uniformly within a predefined range.

A.4.8 MAXIMUM CLIQUE PROBLEM

The Maximum Clique Problem (MCP) is defined on an undirected graph $G = (V, E)$. A clique is a subset of vertices $C \subseteq V$ such that every pair of distinct vertices in $C$ is connected by an edge in $E$. The problem asks for the largest such subset, i.e., a clique of maximum cardinality.

The solution is expressed as a list of vertex IDs forming the clique. For example, given the adjacency lists

$$0 : [1, 2, 3, 4], \quad 1 : [0, 3, 4], \quad 2 : [0, 3], \quad 3 : [0, 1, 2, 4], \quad 4 : [0, 1, 3],$$

a valid maximum clique is

$$[0, 1, 3, 4],$$

which has size 4.

The difficulty of generated MCP instances depends on the graph size and density:

- **Easy**: 4–8 vertices, cliques of size 2–4.
- **Medium**: 8–12 vertices, cliques of size 2–4.
- **Hard**: 12–16 vertices, cliques of size 2–6.
- **Benchmark**: 16–20 vertices, cliques of size 4–8.

Graphs are generated by first constructing a guaranteed clique and embedding it into a larger graph with random edges, ensuring the clique exists as the maximum solution.

A.4.9 MAXIMUM INDEPENDENT SET

The Maximum Independent Set (MIS) problem is defined on an undirected graph $G = (V, E)$. An independent set is a subset of vertices $I \subseteq V$ such that no two vertices in $I$ are adjacent in $G$. The problem asks for the independent set of maximum cardinality.

The solution is expressed as a list of vertex IDs forming the set. For example, given the adjacency lists

$$0 : \{1, 2\}, \quad 1 : \{0, 2, 3\}, \quad 2 : \{0, 1, 3\}, \quad 3 : \{1, 2\},$$

a maximum independent set is

$$[0, 3],$$

which has size 2.

The difficulty of generated MIS instances depends mainly on the graph size and the planted independent set:

- **Easy**: 12–20 vertices, independent set size 4–8.
- **Medium**: 20–30 vertices, independent set size 8–12.
- **Hard**: 30–40 vertices, independent set size 12–16.
- **Benchmark**: 40–50 vertices, independent set size 16–20.

Graphs are generated by first selecting a guaranteed independent set and embedding it into a larger graph with randomly added edges, ensuring the independent set exists as the maximum solution.

A.4.10 GRAPH COLORING PROBLEM

The Graph Coloring Problem (GCP) is defined on an undirected graph $G = (V, E)$. The task is to assign a color to each vertex such that no two adjacent vertices share the same color, while minimizing the total number of colors used.

The solution is expressed as a list of integers, where the $i$-th entry denotes the color assigned to vertex $i$. For example, given the adjacency lists

$$0 : [1, 2], \quad 1 : [0, 3], \quad 2 : [0, 3], \quad 3 : [1, 2],$$

a valid optimal coloring is

$$[1, 2, 1, 2],$$

which uses 2 colors.

The difficulty of generated GCP instances depends mainly on the number of vertices, the number of colors required, and the edge density:

- **Easy**: 8–12 vertices, 3–4 colors, edge density $\approx 0.2$.
- **Medium**: 15–22 vertices, 4–6 colors, edge density $\approx 0.35$.
- **Hard**: 25–32 vertices, 6–8 colors, edge density $\approx 0.5$.
- **Benchmark**: 32–40 vertices, 6–8 colors, edge density $\approx 0.5$.

Graphs are generated by partitioning vertices into color classes and adding random edges between different partitions, ensuring that the planted coloring remains a valid optimal solution.

