# OpenReview forum: "NP-ENGINE: EMPOWERING OPTIMIZATION REASON- ING IN LARGE LANGUAGE MODELS WITH VERIFIABLE SYNTHETIC NP PROBLEMS"
_ICLR.cc/2026/Conference — ICLR 2026 Conference Withdrawn Submission_

### Official Review · Reviewer_YXAd · 2025-10-16

**Soundness:** 3
**Presentation:** 2
**Contribution:** 2
**Rating:** 4
**Confidence:** 4

**Summary:**

The paper introduces NP-ENGINE, a framework designed to enhance the optimization reasoning capabilities of Large Language Models (LLMs) using Reinforcement Learning with Verifiable Rewards (RLVR). The framework features a pipeline to generate instances of 10 NP-hard problems, verify solutions automatically, and use heuristic solvers to provide reward signals. Using this system, the authors create the NP-BENCH benchmark and train a 7-billion parameter model, QWEN2.5-7B-NP, demonstrating significant performance improvements on in-domain tasks.

**Strengths:**

1) The paper tackles the important area LLM optimization, moving beyond simple binary correctness to evaluate and improve the quality of solutions for combinatorial problems.

2) Results demonstrate clear, significant performance gains on NP-BENCH, showing that RLVR with heuristic supervision can improve structured combinatorial reasoning.

3) The generator–verifier–heuristic setup and hierarchical difficulty scaling are well-motivated for RLVR training.

**Weaknesses:**

1) The paper calls NP-ENGINE “the first comprehensive framework for NP-hard training and evaluation”, but efforts with significant overlap pre-date it:
- Reasoning Gym (2025) is explicitly a library of RL with verifiable rewards—100+ generators and verifiers across math/logic/graph/game domains, many of which are combinatorial problems.These clearly establish prior art.
-NPHardEval (ACL 2024): a dynamic NP-class benchmark refreshed monthly of ~ 900 problems spanning P, NP, and NP-hard complexity levels, with automatic verifiers and difficulty control.
- NPPC (Nondeterministic Polynomial-time Problem Challenge, 2025): exposed 25 NP-complete tasks with modules that generate, solve, and evaluate instances across scaling dimensions.
- SearchBench provides instance generators with difficulty control and automated pipelines to evaluate feasibility, correctness, and optimality, and which can supply the verifiable reward signals needed for RLVR.

2) The paper compares a domain-specialized, RLVR-trained Qwen2.5-7B-NP to general-purpose baselines (e.g., GPT-4o, DeepSeek R1). This shows in-domain adaptation, not necessarily improved ability to solve combinatorial problems. Fair evaluation should test transfer to other combinatorial benchmarks (e.g., NPPC, NPHardEva, SearchBenchl) to measure improvement in optimization reasoning.

3) The NP-ENGINE generators are described qualitatively but not referenced with external verifiable sources. There is no proof or diagnostic provided ensuring that generated instances avoid trivial subclasses or guarantee solvable instances.

4) It’s not demonstrated whether RLVR training generalizes across model architectures or comparable-size models (e.g., LLaMA, Mistral). If compute is a bottleneck, the authors could include ablations on similarly sized or smaller models for comparison.

5) The main result table (Table 2) is overloaded with data and poorly structured, making them difficult to parse and reducing the paper’s overall readability.

**Questions:**

1) Can you please define “trainable” in Table 1? Given that SearchBench already includes generators and feasibility/optimality verifiers, what prevents its direct use for training/RLVR?

2) Considering the significant  overlap with prior work (e.g., NPHardEval's procedural generation, NPPC's RL-ready environments, Reasoning gym), could you clarify the key novel contributions of the NP-ENGINE framework?

4) How do you guarantee that generated NP instances are solvable or non-trivial?

---

### Official Review · Reviewer_BuLi · 2025-10-28

**Soundness:** 2
**Presentation:** 1
**Contribution:** 1
**Rating:** 2
**Confidence:** 4

**Summary:**

The paper presents NP-Engine, a task suite for NP-hard optimization problems equipped with difficulty-adjustable data generators along with formal solvers and validators. The paper also sets a reward scheme for finetuning models with RLVR on optimization problems. The authors evaluate several pretrained LLM baselines on their benchmark (NP-Bench), and report the performance resulting from finetuning one of them (Qwen2.5-7B) with RLVR on the NP-Bench tasks. The authors further evaluate the impact of their training pipeline on the Qwen2.5-7B performance on out-of-domain reasoning benchmarks, where they find that pretraining Qwen2.5-7B on NP-Engine tasks improves model performance.

**Strengths:**

1. The experimental volume is fairly large.
2. The presented NP-Bench benchmark could be useful for assessing LLM performance on these optimization problems in future research.

**Weaknesses:**

1. This paper belongs more in the datasets and benchmarks track since its main contribution revolves around task benchmarking.
2. Even though the authors evaluate several proprietary and open-source LLMs on their benchmark (reported in Tables 2 & 3), they apply their training pipeline to only one of them, and one of the worst-performing at that. This is rather strange; why not apply it to the best-performing base LLM if the goal is to advance the state-of-the-art? Ideally, the training pipeline should be evaluated on all of them.
3. The performance improvements in the last row of Table 2 are quite unsurprising: one would naturally expect that specifically finetuning a model (Qwen2.5-7B) for the NP-Bench tasks would make them perform much better on these very tasks. For the same reason, it is not striking that the finetuned LLM outperforms GPT-4o on several NP-Bench tasks.
4. The comment in lines 417-418: "As shown in Figure 3, compared to OneStage-RL, which uses NP-ENGINE-DATA in a single epoch, MultiStage-RL divides NP-ENGINE-DATA into multiple epochs." is incorrectly stated: that is not what Figure 3 shows -- it shows the performance comparison, and nothing about the multi-epoch division.
5. The claim in lines 418-420: "The NP-MultiStage-RL strategy consistently outperforms NP-OneStage-RL across all sub-tasks in both in-domain and out-of-domain (OOD) settings." in section 4.3.2 does not match the OOD results in Figure 3, where NP-OneStage-RL, contrary to the claim, actually outperforms NP-MultiStage-RL in KORBench, GPQA, and IF-EVAL while matching it on OlpBench.
6. The first contribution states that the paper introduces a "scalable framework that generates near-infinite and hierarchically difficult NP-hard problems": what does near-infinite here mean? A quantity is either infinite or simply finite. Does "near-infinite" refer to the number of generated instances? If so, then how are they not infinite, given they are produced by parameterized auto-generators?
7. The answer structure for the format reward in Section 3.1 is far too simplistic to "encourage deep reasoning thinking" as stated in Section 3.1. It simply encourages the model to adhere to the syntactic format of the final answer, which is rather easy and requires hardly any reasoning whatsoever.
8. The overall vision of the paper does not address the real fundamental challenges of solving NP-hard problems with neural networks, such as theoretical complexity, completeness, spurious features preventing generalization, etc. Scaling up data generation and supervision, hoping it would overcome these limitations through sheer training volume, is misguided.

**Questions:**

1. Can you evaluate your training pipeline on one of the better-performing base LLMs, such as Qwen3-32B, for both in-domain and out-of-domain tasks (OOD being more important)? If time and resources permit, it would be interesting to see the results on all base LLMs or at least the best performers.
2. The format and feasibility rewards overlap in the case of an incorrect-format solution since a reward in the incorrect format would also be invalid/infeasible, right? If so, why not set the infeasible-solution reward to the format reward and make it the only reward component?
3. Table 5 reports the performance gains as a result of pretraining on different numbers of tasks from NP-Bench. The results (and intuition) suggest that it is not the count that matters but rather the specific task selection that relates to the OOD tasks. Can you report the performance confusion matrix of pretraining tasks against target OOD tasks? It would be insightful to know which NP-Hard tasks are most relevant/useful to the OOD benchmark tasks.

---

### Official Review · Reviewer_aVCs · 2025-10-30

**Soundness:** 3
**Presentation:** 3
**Contribution:** 3
**Rating:** 6
**Confidence:** 4

**Summary:**

This paper deals with reasoning of LLMs for NP-hard problems. Authors propose a framework, named NP-ENGINE that include both training and evaluation covering 10 tasks across five domains. Compared to previous arts, NP-ENGINE includes instance generators/verifiers/solvers that affords scaling to large dataset and supports RLVR training. Along with NP-ENGINE, this paper also introduce a benchmark named NP-BENCH for NP-hard level problem reasoning. To demonstrate the effectiveness of NP-ENGINE, authors trained a 7B model from QWEN2.5 which achieved SOTA on NP-BENCH and improved performance on other datasets (OOD) including logic/math/knowledge/instruction following.

**Strengths:**

1. NP-hard problems are of important interests in the filed of Computer Science, introduction of this topic to LLM post-training/evaluation is appropriate and timely
2. This paper proposes NP-ENGINE in a way that promotes scalability by incorporating instance generator, automatic verifier and heuristic baselines, avoid of human annotations.
3. Strong ablations provided in the paper include both data proportion and the curriculum learning
2. This paper proposes a benchmark that is sufficiently difficult for current LLMs

**Weaknesses:**

1. The paper mainly focuses on metrics (AR/SR), it could be great to include some qualitative analysis of how a model tries to solve the problems before and after training with NP-ENGINE.
2. Also 10 NP-hard tasks is good but not great yet.

**Questions:**

1. Is it possible that in some cases one of the LLMs' answers actually surpasses that of the heuristic baseline?

---

### Official Review · Reviewer_oKAD · 2025-11-03

**Soundness:** 2
**Presentation:** 3
**Contribution:** 3
**Rating:** 4
**Confidence:** 3

**Summary:**

This paper proposes NP-Bench, which includes ten different categories of NP-hard problems, to evaluate the reasoning ability of models. It also conducts RLVR experiments based on this dataset, and the experimental results show that RLVR can significantly improve the model's performance on NP-Bench. Meanwhile, it also has a certain generalization ability for other OOD tasks to a certain extent.

**Strengths:**

1. While the NP-Hard benchmark is not a particularly novel idea, this paper has explored this topic in greater comprehensiveness and depth. It covers the entire scope from the benchmark itself to RLVR, providing very solid experimental results.
2. This paper is well-written. I am enjoying reading this paper. Good job.

**Weaknesses:**

1. **Missing details of RLVR experiment.** I believe the RLVR experiment is one of the most important contributions of this paper. While the RL experiments in this paper have achieved impressive results, they do not seem to provide much descriptions of the RLVR training process and details. Some aspects that audience may care about include the convergence of train score and test score, changes in entropy, and changes in response length during the training process, demo cases, among others. Furthermore, could the authors provide more detailed experimental settings, such as clip ratio? Line 613 mentions that the global batch size for RL is 8, while the mini batch size is 256. Why is the global batch size smaller than the mini batch size? Additionally, this paper conducts training based on Qwen2.5-7B-Instruct-1M, so it would be better to take it as the baseline and provide this model's scores in the experimental results.
2. **Reward shaping needs further refinement.** This paper directly uses the ratio of the objective value to the optimal value as the reward for optimality. However, each type of NP-Hard problem has a different distribution in terms of objective values: some tasks may have generally higher overall objective values, meaning even random solutions can achieve decent results; other tasks may have larger overall variances. This is not an issue for GRPO, as GRPO performs reward normalization. But it may lead to bias in the benchmark, potentially causing the weights of certain categories of problems to be amplified.

**Questions:**

Please refer to "Weaknesses".

---

> ### Author Response · Authors · 2025-11-21
>
> Thank you for your insightful comments.
>
> A1: We apologize for the oversight regarding the details of RLVR. Due to space limitations, we inadvertently omitted some key details, which we will promptly include in the revised version. Regarding the clip ratio, we have set it to 0.2, and we apologize for any confusion caused by the discrepancy between the global batch size and mini batch size. In practice, we used a smaller value between the global batch size and mini batch size, specifically a mini batch size of 8 during the RLVR training. We will clarify this aspect in the updated manuscript to avoid further misunderstanding.
>
> A2:   We appreciate the reviewer’s feedback on reward shaping. To mitigate potential bias, we control task difficulty in our RL training data. For the Qwen2.5-7b-ins-1M model, we categorize data into easy, medium, and hard, with success rates set at around 0.4, 0.25, and 0.15, respectively. Additionally, each model solution is verified rigorously, making it difficult for random solutions to receive high rewards.
>   On evaluation progress, we also ensure fair comparisons by categorizing tasks into five distinct groups with consistent scaling within each group. This approach minimizes any bias and ensures a balanced benchmark.
>
>   Thank you again for your careful review and valuable feedback.

---

### Note · Authors · 2026-01-08

I have read and agree with the venue's withdrawal policy on behalf of myself and my co-authors.